# PlugVFL: Robust and IP-Protecting Vertical Federated Learning against Unexpected Quitting of Parties

## Abstract

In federated learning systems, the unexpected quitting of participants is inevitable. Such quittings generally do not incur serious consequences in horizontal federated learning (HFL), but they do bring damage to vertical federated learning (VFL), which is underexplored in previous research. In this paper, we show that there are two major vulnerabilities when passive parties unexpectedly quit in the deployment phase of VFL — severe performance degradation and intellectual property (IP) leakage of the active party's labels. To solve these issues, we design PlugVFL to improve the VFL model's robustness against the unexpected exit of passive parties and protect the active party's IP in the deployment phase simultaneously. We evaluate our framework on multiple datasets against different inference attacks. The results show that PlugVFL effectively maintains model performance after the passive party quits and successfully disguises label information from the passive party's feature extractor, thereby mitigating IP leakage.

## 1 Introduction

Federated learning (FL) (McMahan et al., 2017; Yang et al., 2019b) is a distributed learning method that allows multiple parties to collaboratively train a model without directly sharing their data, thereby preserving their data privacy. FL was initially proposed as Horizontal Federated Learning (HFL) to enable collaborative learning across devices (Sun et al., 2022). In this case, data is "horizontally" split, where the devices share the same feature space but have different samples. Another FL framework is Vertical Federated Learning (VFL), which focuses on scenarios where various parties have data with different feature spaces but share overlapping samples (Liu et al., 2019; 2021a). Different from HFL, VFL is mostly deployed in cross-silo scenarios. Suppose a service provider, referred to as active party, owns data and labels of its clients and wishes to train a deep learning model. The service provider may collaborate with other parties, namely passive parties, that possess different data features of the same clients to boost the model's performance. Instead of explicitly sharing the raw data, the passive parties transmit the extracted representations to the active party for training and inference.

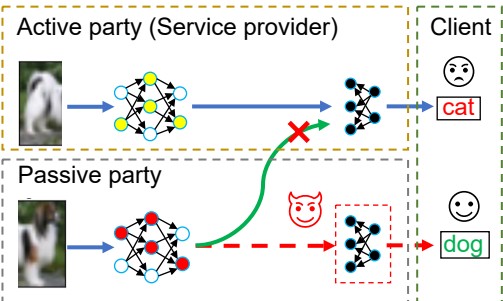

Figure 1: The passive party might quit in the deployment phase, which would cause a substantial performance drop. The passive party could also extract representations containing the information of the active party's labels using its feature extractor, leading to IP leakage of the active party.

The collaboration of devices in HFL happens in the training phase, and the global model is deployed on each device for local inference in the deployment phase. In contrast, VFL requires the parties to collaborate in both the training and deployment phases. During the deployment phase, the active party still requires the representations uploaded by passive parties to conduct inference. However, in real-world scenarios, it is possible for passive parties to quit unexpectedly at inference time due to network crashes, system maintenance, or termination of collaborations. When unexpected quitting happens, the service provider faces two challenges: (1) a substantial **performance drop**; (2) potential **intellectual property (IP) leakage** through the passive party's feature extractor. This paper shows that the drop in model performance caused by the passive party's quitting results in a model that performs worse than one trained by the active party alone, ultimately undermining the motivation for VFL. Furthermore, the passive parties can retain access to their feature extractors even after terminating the collaboration. These feature extractors are trained using the active party's labels, which are valuable IP. From these feature extractors, the passive parties can extract representations containing the information of the active party's labels. Although previous studies made efforts towards mitigating label information leakage through inference attacks on gradients during the training phase (Chaudhuri & Hsu, 2011; Ghazi et al., 2021), the robustness and IP protection of VFL in the deployment phase remain under-explored.

In this paper, we design a framework named PlugVFL to solve the two challenges simultaneously. Specifically, to alleviate the performance drop when passive parties quit unexpectedly, PlugVFL applies an alternative training method that can reduce the co-adaptation of feature extractors across parties. To prevent the IP leakage of the active party's labels, we propose a defense that minimizes the mutual information (MI) between the representations of the passive party and the true labels. We formulate the defense into an adversarial training algorithm that jointly minimizes the variational MI upper bound and prediction loss.

Our key contributions are summarized as follows:

- We reveal two vulnerabilities caused by the unexpected quitting of parties in the deployment phase of VFL, including severe performance drop and active party's label leakage.
- we design a VFL framework PlugVFL to preserve the VFL model's performance against the unexpected exit of passive parties and protect the active party's IP in the deployment phase simultaneously.
- We empirically evaluate the performance of our framework with different datasets. Our results show that PlugVFL can improve the accuracy after the passive party's exit by more than 8% on CIFAR10. PlugVFL also prevents the passive party from fine-tuning a classifier that outperforms random guess levels even using the entire labeled dataset with only less than 2% drop in the VFL model accuracy, outperforming baselines significantly.

## 2 RELATED WORK

### 2.1 VERTICAL FEDERATED LEARNING

Vertical federated learning (VFL) has been an emerging research area. In contrast to (horizontal) federated learning (HFL), VFL adopts a different scheme for data partitioning (Hardy et al., 2017; Yang et al., 2019b). In VFL, different parties will have various parts of the data of an overlapping individual. There has been an amount of research devoted to VFL. Specifically, Hardy et al. (2017) proposes a protocol involving a trusted third party to manage the communication utilizing homomorphic encryption. Others have been following Hardy et al. (2017), where Nock et al. (2018) is working on assessing the protocols and Yang et al. (2019a;c) are focusing on algorithm design concerning optimization. Additionally, VFL algorithms on traditional machine learning, such as tree-boosting (Cheng et al., 2021), gradient boosting (Wu et al., 2020; Fu et al., 2021), random forest (Liu et al., 2020), linear regression (Zhang et al., 2021), and logistic regression (Hu et al., 2019; Liu et al., 2019) are also proposed. Another line of research is working on communication efficiency (Liu et al., 2019), which decreases the communication frequency by leveraging stale gradients on local training. Besides, the assumption of overlapping individuals in VFL among parties produces a challenge for applying VFL in the real world, where FedMVT (Kang et al., 2020) proposes to estimate representations and labels to alleviate the gap.

## 2.2 IP LEAKAGE IN VFL

Intellectual Property (IP) is drawing more and more attention as the rapid growth of commercial deployment of deep learning, especially in federated learning scenarios, whose primary concern is privacy. IP leakage can be divided into data IP leakages, such as deep leakage from gradients (DLG) (Zhu et al., 2019; Zhao et al., 2020), model inversion (Fredrikson et al., 2015) and their variants (Geiping et al., 2020; Jin et al., 2021; Yin et al., 2021; Melis et al., 2019; Jiang et al., 2022), and model IP leakage, such as model extraction attacks (Tramèr et al., 2016; Orekondy et al., 2019a; Pal et al., 2019; Correia-Silva et al., 2018; Truong et al., 2021), where multiple defensive methods have also been proposed to tackle data IP leakage (So et al., 2020; Mo et al., 2021; Abadi et al., 2016; Bonawitz et al., 2017) and model IP leakage (Juuti et al., 2019; Orekondy et al., 2019b).

In VFL, we categorize IP stealing attacks into two types, i.e., feature inference (Luo et al., 2021; He et al., 2019; Jiang et al., 2022; Jin et al., 2021) and label inference (Fu et al., 2022b; Li et al., 2021; Liu et al., 2021b). Specifically, Luo et al. (2021) proposes general attack methods for complex models, such as Neural Networks, by matching the correlation between adversary features and target features, which can be seen as a variant of model inversion (Fredrikson et al., 2015; Sun et al., 2021). He et al. (2019); Jiang et al. (2022) also propose variants of model inversion attack in VFL. While all these attacks are in the inference phase, Jin et al. (2021) proposes a variant of DLG (Zhu et al., 2019) which can perform attacks in the training phase. For label inference, Li et al. (2021) proposes an attack method and a defense method for two-party split learning on binary classification problems, a special VFL setting. Additionally, Fu et al. (2022b) proposes three different label inference attack methods considering different settings in VFL: direct label inference attack, passive label inference attack, and active label inference attack. Defensive methods have also been proposed. For example, Liu et al. (2021b) proposes manipulating the labels following specific rules to defend the direct label inference attack, which can be seen as a variant of label differential privacy (label DP) (Chaudhuri & Hsu, 2011; Ghazi et al., 2021) in VFL. However, all these defending methods focus on preventing data IP leakage from gradients in the training phase. To the best of our knowledge, we are the first to provide an analysis of label IP protection in the VFL deployment phase.

## 3 PROBLEM DEFINITION AND MOTIVATION

### 3.1 VERTICAL FEDERATED LEARNING SETTING

Suppose $K$ parties train a model. There is a dataset[1] across all parties with size $N$: $D = \{x_i, y_i\}_{i=1}^N$. The feature vector $x_i \in \mathbb{R}^d$ is splitted among $K$ parties $\{x_i^k \in \mathbb{R}^{d_k}\}_{k=1}^K$, where $d_k$ is the feature dimension of party $k$, and the labels $Y = \{y_i\}_{i=1}^N$ are owned by one party. The parties with only features are referred to as *passive parties*, and the party with both features and labels is referred to as the *active party*. We denote party 1 as the active party, and other parties are passive parties.

Each party (say the $k$-th) adopts a representation extractor $f_{\theta_k}(\cdot)$ to extract representations of local data $H^k = \{H_i^k\}_{i=1}^N = \{f_{\theta_k}(x_i^k)\}_{i=1}^N$ and sends them to the active party, who possesses labels and a predictor. The overall training objective of VFL is formulated as

$$\min_{\Theta} \mathcal{L}(\Theta; D) \triangleq \frac{1}{N} \sum_{i=1}^N \mathcal{L}\left(\mathcal{S}_{\theta_{\mathcal{S}}}\left(H_i^1, ..., H_i^K\right), y_i\right), \tag{1}$$

where $\Theta = [\theta_1; ...; \theta_K; \theta_{\mathcal{S}}]$, $\mathcal{S}$ denotes a trainable head model on active party to conduct classification, and $\mathcal{L}$ denotes the loss function. The objective of each passive party $k$ is to find an optimal $\theta_k^*$ while not sharing local data $\{x_i^k\}_{i=1}^N$ and parameters $\theta_k$. The objective of the active party is to optimize $\theta_1$ and $\theta_{\mathcal{S}}$ while not sharing $\theta_1$, $\theta_{\mathcal{S}}$ and true labels $Y$. The active party calculates the gradients of received representations and send $\{\frac{\partial \mathcal{L}}{\partial H^k}\}_{k \in [2,...,K]}$ back to passive parties.

Notably, the passive parties still have to communicate with the active party during the inference phase. For a new data $x_i$, the passive parties send the extracted representations $\{H_i^k\}_{k \in [2,...,K]}$ to the active party, and the active party generates the prediction $\mathcal{S}_{\theta_{\mathcal{S}}}\left(H_i^1, ..., H_i^K\right)$.

---

[1]We assume the alignment between overlapping samples is known as a prior. In some applications, private set intersection could be used before running VFL to find the sample alignment.

Table 1: Results before and after party 2 quits on CIFAR10.

| | Accuracy(%) |
|---|---|
| Before party 2 quits | 74.53 |
| After party 2 quits | 51.24 |
| Party 1 standalone | 62.84 |

Table 2: Accuracy of the model on party 2 by conducting MC attack and collecting labels to train from scratch.

| | Accuracy(%) |
|---|---|
| MC attack (400 labels) | 58.02 |
| Train from scratch w. all the labels | 59.73 |

## 3.2 PERFORMANCE DROP AFTER PARTIES QUIT

During the deployment phase, some passive parties (say the $k$-th party) could quit unexpectedly due to a network crash or the termination of collaboration. Without the representations uploaded by party $k$, the active party can still conduct inference by setting $H_i^k$ as a zero vector. However, there will be a substantial performance drop. We conduct two-party experiments on CIFAR10 to investigate this performance drop. We follow previous works (Liu et al., 2019; Kang et al., 2022) to split CIFAR10 images into two parts and assign them to the two parties using ResNet18 as backbone models. The active party (party 1) and passive party (party 2) collaborate to train the models. We evaluate and compare the inference accuracy before and after party 2 quits in the deployment phase. When party 2 quits, party 1 sets $H_i^2$ as a zero vector and conducts inference. Zero vectors are used because the passive party typically does not allow the active party to utilize its representations in any way (e.g., an average vector) after the termination of collaboration. We set the standalone results as a baseline, where the active party trains a model independently without ever collaborating with the passive party.

The results (shown in Tab. 1) demonstrate that the accuracy drops more than 20% after party 2 quits. Furthermore, the VFL model after party 2 quits achieves even lower accuracy than the model party 1 trained without any collaboration, undermining the motivation of VFL.

## 3.3 IP LEAKAGE OF LABELS IN THE DEPLOYMENT PHASE

The collaborative training process enables passive parties to extract representations useful for the task of VFL, which is learned from the labels of the active party. Even after the collaboration ends, the passive parties will retain access to the representation extractors. These extractors allow the passive parties to fine-tune classifier heads with very few labeled data and conduct inference with decent accuracy after quitting the collaboration. Given the active party's significant investment of effort and money in labeling the data, these extractors retained by the passive parties constitute costly IP leakage of these labels. To demonstrate the extent of IP leakage by the feature extractors of the passive parties, we follow the experimental setup in Sec. 3.2 and let party 2 conduct model completion (MC) attack (Fu et al., 2022a) to train a classifier using a small number of labeled samples. We report the test accuracy of the complete model of party 2 created by the MC attack. For comparison, we also assume party 2 annotates all the training data to train a model from scratch.

We report the accuracy in Tab. 2. By fine-tuning a classifier with the extractor, the passive party can achieve comparable accuracy using less than 1% of the labeled data compared to training a model from scratch with all the labels. This demonstrates that the label information from the active party is leaked and embedded in the passive party's extractor.

# 4 METHOD

## 4.1 OVERVIEW OF PLUGVFL

Without loss of generality, we formulate our PlugVFL framework in the two-party scenario. Suppose the passive party (party 2) and active party (party 1) have sample pairs $\left\{ \left( x_i^1, x_i^2, y_i \right) \right\}_{i=1}^N$ drawn from a distribution $p\left( x^1, x^2, y \right)$, and the representations of party $k$ is calculated as $h^k = f_{\theta_k}(x^k)$. We use $h^k$, $x^k$ and $y$ here to represent random variables, while $H_i^k$, $x_i^k$ and $y_i$ stand for deterministic values. Then the training of our framework is to achieve three goals:

- Goal 1: To preserve the performance of VFL, the main objective loss should be minimized.

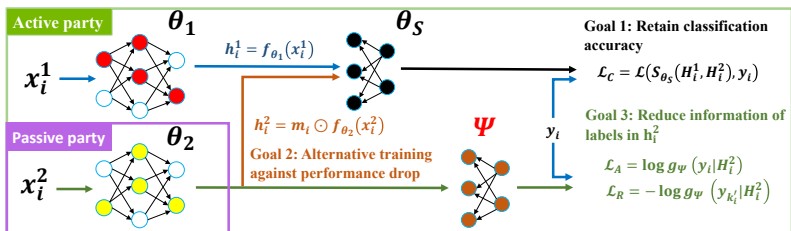

Figure 2: Overview of PlugVFL: The active party conducts alternative training to preserve the performance against the exit of the passive party. IP of labels is protected by optimizing $\mathcal{L}_A$ and $\mathcal{L}_R$ to reduce label information in passive parties' representations.

- Goal 2: To preserve the performance after party 2 quits, the objective loss without the representations of party 2 should be minimized.
- Goal 3: To reduce the IP leakage of labels from party 2, $\theta_2$ should not be able to extract representations $h^2$ containing much information about the true label $y$.

Formally, we have three training objectives:

$$\textbf{Prediction performance: } \min_{\theta_1,\theta_2,\theta_S} \mathcal{L}\left(\mathcal{S}_{\theta_S}\left(h^1,h^2\right),y\right),$$

$$\textbf{Robustness against quitting: } \min_{\theta_1,\theta_2,\theta_S} \mathcal{L}\left(\mathcal{S}_{\theta_S}\left(h^1,0\right),y\right),\tag{2}$$

$$\textbf{Label IP protection: } \min_{\theta_2} \mathrm{I}(h^2;y),$$

where $\mathcal{S}_{\theta_S}\left(h^1,0\right)$ is the prediction when the server does not receive the representations $h^2$ uploaded by party 2. $\mathrm{I}(h^2;y)$ is the mutual information between $h^2$ and $y$, which indicates the information $h^2$ preserves for the label variable $y$. We minimize this mutual information to protect the active party's labels' IP from being steal by the passive party.

## 4.2 Efficient alternative training to Achieve Robustness

A trivial way to improve the robustness against quitting is to combine the training objectives $\mathcal{L}\left(\mathcal{S}_{\theta_S}\left(h^1,h^2\right),y\right)$ and $\mathcal{L}\left(\mathcal{S}_{\theta_S}\left(h^1,0\right),y\right)$ on the server. However, the server has to conduct training of $\mathcal{S}_{\theta_S}$ twice, which involves computational overhead. To improve the efficiency, we propose an alternative training method to achieve the second goal. Specifically, for each communication round (i.e., an iteration of training in VFL), the active party omits the representations from the passive party with probability $p$. The expectation of the training objective is formulated as

$$\mathbb{E}_p \mathcal{L}(\Theta;D) = (1-p)\mathcal{L}\left(\mathcal{S}_{\theta_S}\left(h^1,h^2\right),y\right) + p\mathcal{L}\left(\mathcal{S}_{\theta_S}\left(h^1,0\right),y\right),\tag{3}$$

which is a weighted sum of the first and the second goal with weight $p$. Notably, a larger $p$ sets a larger weight for $\mathcal{L}\left(\mathcal{S}_{\theta_S}\left(h^1,0\right),y\right)$. Thus, $p$ can be chosen based on the chance that party 2 quits.

The intuition behind alternative training is to reduce the co-adaptation between the head predictor and local extractors. The severe performance drop after the quitting of passive parties comes from the co-adaptation of the hidden neurons of the head predictor $\mathcal{S}_{\theta_S}$ and the neurons of local extractors $f_{\theta_k}$, where a hidden neuron of the predictor $\mathcal{S}_{\theta_S}$ only depends on the pattern of several specific neurons of specific parties' extractors. The *dropout* was proposed as an effective solution to co-adaptation (Baldi & Sadowski, 2013). Similar to *dropout*, which omits some neurons, our proposed alternative training method omits the passive party, which solves the party-wise co-adaptation in VFL.

## 4.3 Variational Training Objective of Label Protection

The mutual information term (i.e., goal 3) is hard to compute in practice as the random variable $h^2$ is high-dimensional. In addition, computing mutual information requires knowing the distribution $p(y|h^2)$, which is difficult to obtain. To derive a tractable estimation of the mutual information objective, we leverage *CLUB* (Cheng et al., 2020) to formulate a variational upper-bound:

$$\mathrm{I}\left(h^2;y\right) \leq \mathrm{I}_{\mathrm{vCLUB}}\left(h^2;y\right) := \mathbb{E}_{p\left(h^2,y\right)}\log q_\psi\left(y|h^2\right) - \mathbb{E}_{p\left(h^2\right)p(y)}\log q_\psi\left(y|h^2\right),\tag{4}$$

**Algorithm 1 Training algorithm of PlugVFL.** ← means information is sent to the active party; ← means information is sent to the passive party; red steps are conducted on the passive party.

**Input:** Dataset $\left\{\left(x_i^1, x_i^2, y_i\right)\right\}_{i=1}^N$; Learning rate $\eta$.
**Output:** $\theta_1; \theta_{\mathcal{S}}; \psi$.
1: Initialize $\theta_1; \theta_{\mathcal{S}}; \psi$;
2: **for** a batch of data $\left\{\left(x_i^1, x_i^2, y_i\right)\right\}_{i \in \mathbb{B}}$ **do**
3: $\quad \{H_i^2\}_{i \in \mathbb{B}} \leftarrow \{f_{\theta_2}\left(x_i^2\right)\}_{i \in \mathbb{B}}$;
4: $\quad \mathcal{L}_A \leftarrow \frac{1}{|\mathbb{B}|} \sum_{i \in \mathbb{B}} \log g_\psi\left(y_i | H_i^2\right)$;
5: $\quad \psi \leftarrow \psi + \eta \nabla_\psi \mathcal{L}_A$;
6: $\quad$ Randomly generate binary mask vectors $\{m_i\}$ with probability $p$ to be zeros;
7: $\quad \mathcal{L}_C \leftarrow$
$\quad\quad \frac{1}{|\mathbb{B}|} \sum_{i \in \mathbb{B}} \mathcal{L}\left(\mathcal{S}_{\theta_{\mathcal{S}}}\left(f_{\theta_1}\left(x_i^1\right), m_i \odot H_i^2\right), y_i\right)$;

8: $\quad \theta_1 \leftarrow \theta_1 - \eta \nabla_{\theta_1} \mathcal{L}_C$;
9: $\quad \theta_{\mathcal{S}} \leftarrow \theta_{\mathcal{S}} - \eta \nabla_{\theta_{\mathcal{S}}} \mathcal{L}_C$;
10: $\quad \{y_{k_i'}\}_{i \in \mathbb{B}} \leftarrow$ randomly sample $\{y_{k_i'}\}_{i \in \mathbb{B}}$ from $\{y_i\}_{i \in [N]}$.
11: $\quad \mathcal{L}_R \leftarrow \frac{1}{|\mathbb{B}|} \sum_{i \in \mathbb{B}} -\log g_\psi\left(y_{k_i'} | H_i^2\right)$;
12: $\quad \{\nabla_{H_i^2} \mathcal{L}\}_{i \in \mathbb{B}} \leftarrow$
$\quad\quad \{\nabla_{H_i^2}\left[(1-\lambda)\mathcal{L}_C + \lambda(\mathcal{L}_A + \mathcal{L}_R)\right]\}_{i \in \mathbb{B}}$;
13: $\quad \nabla_{\theta_2} \mathcal{L} \leftarrow \frac{1}{|\mathbb{B}|} \sum_{i \in \mathbb{B}} \nabla_{H_i^2} \mathcal{L} \nabla_{\theta_2} H_i^2$
14: $\quad \theta_2 \leftarrow \theta_2 - \eta \nabla_{\theta_2} \mathcal{L}$;
15: **end for**

where $q_\psi\left(y|h^2\right)$ is a variational distribution with parameters $\psi$ to approximate $p\left(y|h^2\right)$. To reduce the computational overhead, we apply vCLUB-S MI estimator (Cheng et al., 2020), which is an unbiased estimator of $I_{vCLUB}$. The label IP protection objective is equivalent to the following:

$$\min_{\theta_2} I(h^2; y) \Leftrightarrow \min_{\theta_2} \hat{I}_{vCLUB\text{-}S}(h^2; y) = \min_{\theta_2} \frac{1}{N} \sum_{i=1}^N \left[\max_\psi \log q_\psi\left(y_i | H_i^2\right) - \log q_\psi\left(y_{k_i'} | H_i^2\right)\right]. \quad (5)$$

The details can be found in Apendix B.

### 4.4 TRAINING PROCEDURE

We use $g_\psi$ to parameterize $q_\psi$ in Eq. (5). By combining Eq. (5) and the prediction objective with alternative training using a hyperparameter $\lambda$, the overall objective has three terms. The first term is the prediction objective with alternative training, denoted as $\mathcal{L}_C^i = \mathcal{L}\left(\mathcal{S}_{\theta_{\mathcal{S}}}\left(f_{\theta_1}\left(x_i^1\right), m_i \odot f_{\theta_2}\left(x_i^2\right)\right), y_i\right)$, where $m_i$ is a binary vector whose elements will be set as 0's with the probability of $p$. The second term is an adversarial training objective, where an auxiliary predictor $g_\psi$ is trained to capture label information while the feature extractor $f_{\theta_2}$ is trained to extract as little label information as possible, denoted as $\mathcal{L}_A^i = -CE\left(g_\psi\left(f_{\theta_2}\left(x_i^2\right)\right), y_i\right)$. The third term regularizes $f_{\theta_2}$ to capture the information of a randomly selected label, denoted as $\mathcal{L}_R^i = CE\left(g_\psi\left(f_{\theta_2}\left(x_i^2\right)\right), y_{k_i'}\right)$. The loss $\mathcal{L}_A$ and $\mathcal{L}_R$ are formulated from Eq. (5). We reorganize the overall training objective as:

$$\theta_1, \theta_2, \theta_{\mathcal{S}}, \psi = \arg\min \frac{1}{N} \sum_{i=1}^N \min_{\theta_2} \left[(1-\lambda) \min_{\theta_1, \theta_{\mathcal{S}}} \mathcal{L}_C^i + \lambda \max_\psi \mathcal{L}_A^i + \lambda \mathcal{L}_R^i\right]. \quad (6)$$

Our algorithm is summarized in Alg. 1. For each batch of data, we first optimize $\theta_1$ and $\theta_{\mathcal{S}}$ based on the primary task loss. Then we optimize the auxiliary predictor $\psi$. Finally, $\theta_2$ is optimized with $(1-\lambda)\mathcal{L}_C + \lambda\mathcal{L}_A + \lambda\mathcal{L}_R$. Note that $\theta_1, \theta_{\mathcal{S}}$ and $\psi$ are owned by the active party, and their optimization does not require additional information from the passive party except the representations $h^2$, which are uploaded to the active party even without defense. For the passive party, the training procedure of local extractor $\theta_2$ does not change, making our defense concealed from the passive party.

## 5 EXPERIMENTS

We evaluate our proposed PlugVFL on multiple datasets. We focus on two-party scenarios following the VFL literature (Fu et al., 2022b; Liu et al., 2019; 2020; 2021a; Yang et al., 2019b).

**Baselines.** To thoroughly analyze PlugVFL, we first evaluate PlugVFL against performance drop and label leakage separately and compare with the baselines achieving the same goal, respectively. To our knowledge, PlugVFL is the first approach to mitigate the performance drop after the passive party quits in VFL. But we still compare with a baseline, where the active party trains an additional head model without the quitting party to make predictions if the passive party quits, which we call *Multi-head training*. For defense against label leakage, we evaluate PlugVFL against two attacks:

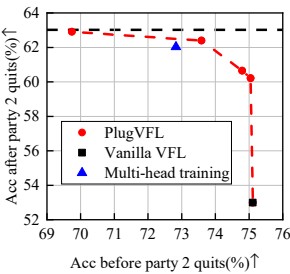

Figure 3: Results of PlugVFL on CIFAR10. The black dashed line denotes the accuracy of the model that party 1 trains independently.

Table 3: Results of PlugVFL on CIFAR100.

|  | Accuracy before party 2 quits(%) | Accuracy after party 2 quits(%) |
|---|---|---|
| $p = 0$ | 44.95 | 26.65 |
| $p = 0.05$ | 44.58 | 32.01 |
| $p = 0.1$ | 44.29 | 32.03 |
| $p = 0.3$ | 42.11 | 33.05 |
| $p = 0.5$ | 40.29 | 33.85 |
| Standalone | N/A | 34.02 |
| Multi-head training | 39.17 | 32.72 |

(1) *Passive Model Completion (PMC)* (Fu et al., 2022b) attack assumes that the passive party has access to an auxiliary labeled dataset. The passive party utilizes this auxiliary dataset to fine-tune a classifier that can be applied to its local feature extractor. (2) *Active Model Completion (AMC)* (Fu et al., 2022b) attack is included as an *adaptive attack* method when the passive party is aware of our method. The passive party conducts AMC to trick the federated model to rely more on its feature extractor so as to increase its expressiveness. The passive party conducts AMC attack by actively adapting its local training configurations. We compare PlugVFL with four existing defense baselines: (1) *Noisy Gradient (NG)* (Fu et al., 2022a) is proven effective against privacy leakage in FL by adding Laplacian noise to gradients. (2) *Gradient Compression (GC)* (Fu et al., 2022a) prunes gradients that are below a threshold magnitude, such that only a part of gradients are sent to the passive party. (3) *Privacy-preserving Deep Learning (PPDL)* (Shokri & Shmatikov, 2015) is a comprehensive privacy-enhancing method including three defense strategies: differential privacy, gradient compression, and random selection. (4) *DiscreteSGD (DSGD)* (Fu et al., 2022a) conducts quantization to the gradients sent to the passive party such that the discrete gradients are used to update the adversarial party's extractor.

**Datasets.** We evaluate PlutVFL on CIFAR10 and CIFAR100. We follow Liu et al. (2019); Kang et al. (2022); Liu et al. (2021a); Yang et al. (2019b) to split images into halves.

**Hyperparameter configurations.** For both CIFAR10 and CIFAR100, we use ResNet18 as backbone models with batch size 32. We apply SGD optimizer with learning rate 0.01. We apply a 3-layer MLP to parameterize $g_\psi$ for PlugVFL. For NG defense, we apply Laplacian noise with mean of zero and scale between 0.0001-0.01. For GC baseline, we set the compression rate from 90% to 100%. For PPDL, we set the Laplacian noise with scale of 0.0001-0.01, $\tau = 0.001$ and $\theta$ between 0 and 0.01. For DSGD, we set the number of gradient value's levels from 1 to 2 and added Laplacian noise with the same scale as PPDL. To simulate the realistic settings in that the passive party uses different model architectures to conduct MC attacks, we apply different model architectures (MLP & MLP_sim) for MC attacks. The detailed architectures can be found in Appendix A. The passive party has 40 and 400 labeled samples to conduct MC attacks for CIFAR10 and CIFAR100, respectively.

**Evaluation metrics.** (1) *Utility metric (Model accuracy)*: We use the test data accuracy of the classifier on the active party to measure the performance. (2) *Robustness metric (Attack accuracy)*: We use the test accuracy of the passive party's model after MC attack to evaluate the effectiveness of our IP protecting method. The lower the attack accuracy, the higher the robustness against IP leakage.

## 5.1 RESULTS OF PERFORMANCE PRESERVATION AGAINST UNEXPECTED EXIT

To evaluate the effectiveness of the alternative training against performance drop, we first conduct experiments by setting $\lambda$ as 0 in Eq. (6). We set $p$ from 0 to 0.5 to simulate the settings that the passive party has different levels of reliability. We evaluate the trade-off between the accuracy before and after the passive party quits in the deployment phase. The results of CIFAR10 are shown in Fig. 3, and the results of CIFAR100 are shown in Tab. 3. The upper bound of the test accuracy after party 2 quits is the accuracy of the model that party 1 trains independently (standalone). For CIFAR10, PlugVFL can improve the accuracy after party 2 quits by more than 7% with nearly no accuracy drop before party 2 quits. By applying alternative training, the active party can achieve nearly the same accuracy as retraining a model locally after the passive party quits by sacrificing less than 1.5%

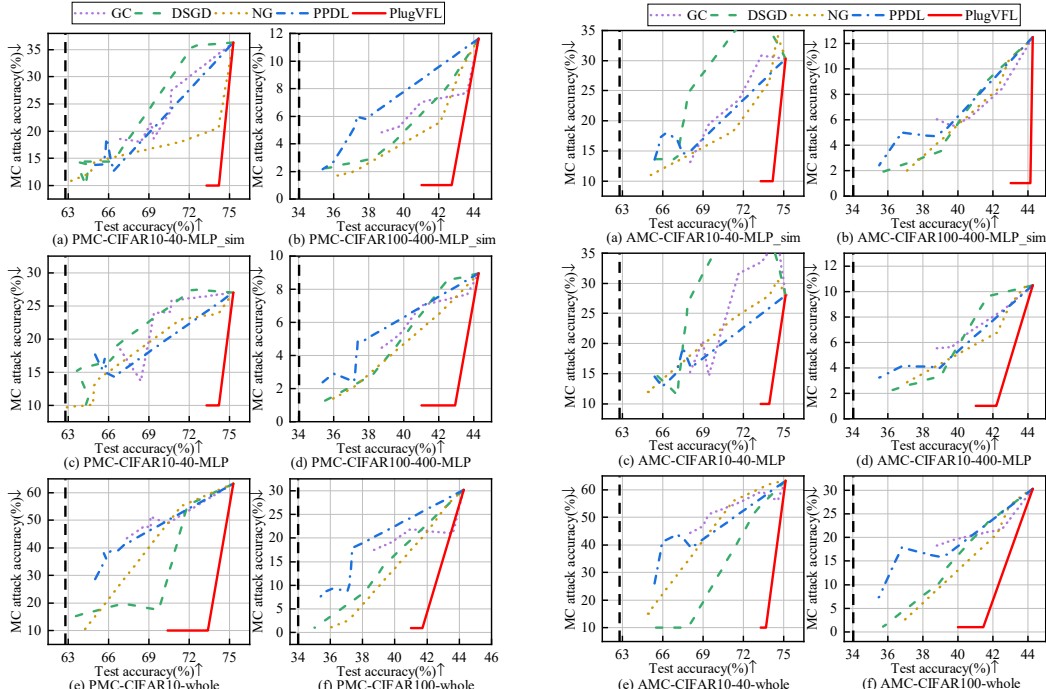

Figure 4: Results of model accuracy v.s. attack accuracy on CIFAR10 and CIFAR100 against PMC attack. The black dashed line denotes the accuracy of that party 1 trains independently.

Figure 5: Results of model accuracy v.s. attack accuracy on CIFAR10 and CIFAR100 against AMC attack. The black dashed line denotes the accuracy of that party 1 trains independently.

accuracy before the passive party quits. Multi-head training can also mitigate the accuracy drop after party 2 quits. However, it cannot achieve a better trade-off than ours since it introduces computational overhead increasing exponentially with $K$.

For CIFAR100, PlugVFL improves the accuracy after party 2 quits by more than 5.5% with less than 0.5% accuracy drop before party 2 quits. It is shown that applying alternative training by just setting a relatively small $p$ value can significantly improve the robustness of VFL against unexpected quitting, demonstrating the effectiveness of PlugVFL in solving the problem of party-wise co-adaptation.

A naïve solution for mitigating the accuracy drop is to fine-tune the head model after a passive party quits. However, this process is time-consuming, and the service provider cannot afford to shut down the service while fine-tuning. Therefore, achieving a decent accuracy before fine-tuning is crucial.

## 5.2 RESULTS OF DEFENSE AGAINST LABEL LEAKAGE

To evaluate the effectiveness of PlugVFL against label IP leakage, we conduct experiments setting $p$ as 0 in alternative training. We evaluate PlugVFL on two datasets against two attack methods. We set different defense levels for our methods (i.e., different $\lambda$ values in Eq. (6)) and baselines to show the trade-off between the model accuracy and attack accuracy. The defense results against PMC and AMC attacks are shown in Fig. 4 and Fig. 5, respectively. To evaluate the effectiveness of our defense in extreme cases, we also conduct experiments that the passive party has the whole labeled dataset to perform MC attacks, of which the results are shown in sub-figures (e) and (f) of Fig. 4 and Fig. 5.

For defense against PMC on CIFAR10, our PlugVFL can achieve 10% attack accuracy (equal to random guess) by sacrificing less than 2% model accuracy, while the other defenses drop model accuracy by more than 12% to achieve the same defense performance. Similarly, our PlugVFL can achieve 1% attack accuracy on CIFAR100 while maintaining a model accuracy drop of less than 3%. In contrast, the other defenses drop model accuracy by more than 9% to achieve the same attack accuracy. Even if the passive party conducts attacks using the whole labeled training dataset, PlugVFL can reduce attack accuracy to random guess with less than 3% model accuracy drop.

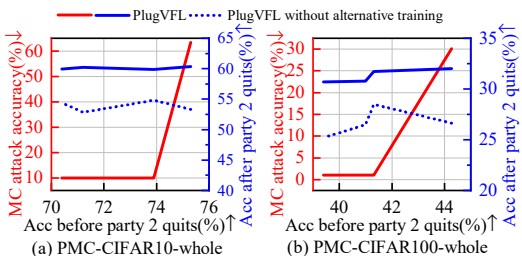
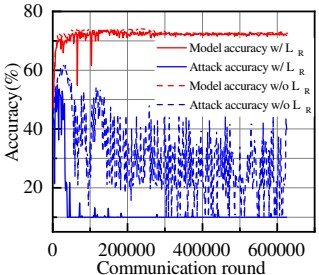

Figure 6: The results of PlugVFL against performance drop and label IP leakage simultaneously.

Figure 7: Defense results on model accuracy and attack accuracy with and without $\mathcal{L}_R$ on CIFAR10 against PMC attack.

Our method achieves similar results against AMC. PlugVFL can achieve high defense performance of an attack accuracy rate of random guess with nearly no model accuracy drop. Notably, the other baselines improve the attack accuracy of AMC in some cases (Fig. 5.(a) and (c)). The reason is that, by applying AMC, the model updating of the passive party is adaptive to the defense methods, making the global classifier rely more on the passive party's feature extractor.

Notably, the baselines achieve low attack accuracy only when the test accuracy degrades to nearly independent training level, that is, baselines can only achieve strong defense performance by severely limiting the expressiveness of the passive party's feature extractor. Our method can achieve a better trade-off between the model utility and the defense performance because PlugVFL only reduces the information of the true labels in the representations extracted by the passive party's feature extractor, while the general information of the data is preserved in the passive party's representations.

### 5.3 RESULTS OF THE INTEGRATED FRAMEWORK

We evaluate PlugVFL against performance drop and label leakage simultaneously under PMC-CIFAR10-whole and PMC-CIFAR100-whole settings. The passive party quits in the deployment phase and tries to conduct a model completion attack using the labeled dataset. We set $p = 0.05$ and $\lambda$ from 0 to 1 for PlugVFL. The results in Fig. 6 show that by applying alternative training, the active party achieves 5% higher accuracy than without alternative training if the passive party quits. Further, we prevent the passive party from achieving an attack accuracy higher than random guess levels using its feature extractor by sacrificing less than 3% model accuracy for both datasets. Thus, our proposed PlugVFL can improve the robustness of VFL against unexpected quitting and protect the active party's label IP effectively.

### 5.4 OBJECTIVE ANALYSIS OF PLUGVFL

The training objective Eq. (6) of PlugVFL consists of 3 terms: $\mathcal{L}_C$, $\mathcal{L}_A$ and $\mathcal{L}_R$. $\mathcal{L}_C$ maintains the model utility. $\mathcal{L}_A$ is the adversarial objective to reduce the information of labels in the passive party's representations. $\mathcal{L}_R$ is also derived from the goal of mutual information reduction, but it is non-trivial to describe its functionality. To analyze the effect of $\mathcal{L}_R$, we conduct experiments that train with and without the objective $\mathcal{L}_R$ under the setting PMC-CIFAR10-whole. The results are shown in Fig. 7. Notably, $\mathcal{L}_R$ does not influence the model accuracy, but the defense performances differ. It is shown that without $\mathcal{L}_R$, the attack accuracy can also degrade to 10% in some communication rounds, but the degradation is much slower than training with $\mathcal{L}_R$. In addition, applying $\mathcal{L}_R$ can stabilize the defense's performance. Thus, $\mathcal{L}_R$ can boost and stabilize the performance of PlugVFL.

### 6 CONCLUSION

We propose a framework PlugVFL that maintains model performance after the passive party quits VFL in the deployment phase and mitigates the active party's label IP leakage simultaneously. The experimental results show that PlugVFL can improve the robustness against unexpected quitting and protect the active party's IP effectively. In this paper, we evaluate the two-party scenario, but our theory and algorithm are naturally extendable to settings with more parties.

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

## A    MODEL ARCHITECTURE

The passive party uses models with different architectures (MLP and MLP_sim) to conduct MC attacks. MLP_sim has one FC layer. MLP has three FC layers with a hidden layer of size $512 \times 256$.

## B    VARIATIONAL TRAINING OBJECTIVE OF LABEL PROTECTION

The prediction objective is usually easy to calculate (e.g., cross-entropy loss for classification). The mutual information term (i.e., goal 3) is hard to compute in practice as the random variable $h^2$ is high-dimensional. In addition, the mutual information requires knowing the distribution $p(y|h^2)$, which is difficult to compute. To derive a tractable estimation of the mutual information objective, we leverage *CLUB* (Cheng et al., 2020) to formulate a variational upper-bound:

$$
\begin{aligned}
&\mathrm{I}\left(h^2; y\right) \\
\leq\; &\mathrm{I}_{\text{vCLUB}}\left(h^2; y\right) \\
:=\; &\mathbb{E}_{p\left(h^2, y\right)} \log q_\psi\left(y|h^2\right) - \mathbb{E}_{p\left(h^2\right) p(y)} \log q_\psi\left(y|h^2\right),
\end{aligned}
\tag{7}
$$

where $q_\psi\left(y|h^2\right)$ is a variational distribution with parameters $\psi$ to approximate $p\left(y|h^2\right)$. To reduce the computational overhead of the defense, we apply the sampled vCLUB (vCLUB-S) MI estimator in Cheng et al. (2020), which is an unbiased estimator of $\mathrm{I}_{\text{vCLUB}}$ and is formulated as

$$
\hat{\mathrm{I}}_{\text{vCLUB-S}}(h^2; y) = \frac{1}{N} \sum_{i=1}^{N} \left[ \log q_\psi\left(y_i|H_i^2\right) - \log q_\psi\left(y_{k_i'}|H_i^2\right) \right],
\tag{8}
$$

where $k_i'$ is uniformly sampled from indices $\{1, ..., N\}$. It is notable that to guarantee the first inequality of Eq. (7), $q_\psi\left(y|h^2\right)$ should satisfy

$$
\mathrm{KL}\left(p\left(h^2, y\right) || q_\psi\left(h^2, y\right)\right) \leq \mathrm{KL}\left(p\left(h^2\right) p(y) || q_\psi\left(h^2, y\right)\right),
\tag{9}
$$

which can be achieved by minimizing $\mathrm{KL}\left(p\left(h^2, y\right) || q_\psi\left(h^2, y\right)\right)$:

$$
\begin{aligned}
&\min_\psi \mathrm{KL}\left(p\left(h^2, y\right) || q_\psi\left(h^2, y\right)\right) \\
=\; &\min_\psi \mathbb{E}_{p\left(h^2, y\right)} \left[\log\left(p\left(y|h^2\right) p\left(h^2\right)\right) - \log\left(q_\psi\left(y|h^2\right) p\left(h^2\right)\right)\right] \\
=\; &\min_\psi \mathbb{E}_{p\left(h^2, y\right)} \left[\log\left(p\left(y|h^2\right)\right) - \log\left(q_\psi\left(y|h^2\right)\right)\right].
\end{aligned}
\tag{10}
$$

Since the first term has no relation to $\psi$, we just need to minimize $\mathbb{E}_{p(h^2, y)} - \log\left(q_\psi\left(y|h^2\right)\right)$. With samples $\left\{\left(x_i^1, x_i^2, y_i\right)\right\}_{i=0}^{N}$, we can derive an unbiased estimation

$$
\max_\psi \frac{1}{N} \sum_{i=1}^{N} \log q_\psi\left(y_i|H_i^2\right).
\tag{11}
$$

With Eq. (7), Eq. (8) and Eq. (11), the objective of label IP protection can be achieved by optimizing

$$
\begin{aligned}
&\min_{\theta_2} \mathrm{I}(h^2; y) \\
\Leftrightarrow\; &\min_{\theta_2} \hat{\mathrm{I}}_{\text{vCLUB-S}}(h^2; y) \\
=\; &\min_{\theta_2} \frac{1}{N} \sum_{i=1}^{N} \left[\max_\psi \log q_\psi\left(y_i|H_i^2\right) - \log q_\psi\left(y_{k_i'}|H_i^2\right)\right].
\end{aligned}
\tag{12}
$$

