# OpenReview forum: "PlugVFL: Robust and IP-Protecting Vertical Federated Learning against Unexpected Quitting of Parties"
_ICLR.cc/2024/Conference — Submitted to ICLR 2024_

### Official Review · Reviewer_ccWW · 2023-10-14

**Soundness:** 3 good
**Presentation:** 3 good
**Contribution:** 3 good
**Rating:** 6
**Confidence:** 3

**Summary:**

The authors study two issues in two-party VFL that occur when the passive party quits: a) the inference accuracy drops significantly as in VFL the passive party's participation is also required during inference, and b) the passive party might try to extract sensitive information about the active party's labels from the representation extractors. The authors propose an alternative VFL training approach to mitigate both issues: a) with a certain probability p during training, the active party will be set the passive party's representations to 0, and b) the mutual information between labels and representations is minimized. An evaluation of ResNet18 training on CIFAR10/100 clearly demonstrates the issues without the proposed mitigations and how those mitigations are indeed effective.

**Strengths:**

The paper studies two clearly relevant issues in the deployment of VFL.

The proposed robustness solution is simple yet effective.

The conducted evaluation answers most questions one might have in terms of dependence on parameter selection and comparison to related works.

**Weaknesses:**

The IP leakage issue of VFL that is discussed (the passive party can try to infer information about the active party's labels) is presented in the setting of the passive party quitting unexpectedly. However, to my understanding, this issue is completely unrelated to drop-outs and the same attack could also be carried out in case the passive party still participates in the system.

In §5.1, the authors briefly discuss a "naive" alternative to their approach of ensuring robustness in case of the second party quitting. This alternative, which is described as fine-tuning the head model after the passive party quits, is dismissed as time-consuming and impractical. However, it would be interesting to see how many training iterations are actually necessary (when shifting all training iterations where the passive party's representation is set to 0 till after the point when the passive party quits) until the model reaches accuracy that is somewhat similar to the case where the proposed mitigation is applied; especially when using small p, the required time to take the service offline might be very small.

The work is somewhat limited in studying only a two-party setting where the accuracy drop as a direct consequence of the only other party quitting is obviously the strongest. This is somewhat fine since also many related works on VFL are restricted to the two-party case, which is also realistic in real-world settings. Nevertheless, a discussion on the likely impact of unexpected quitting of one party in a, e.g., five-party scenario would be appreciated.

In Figure 1, it is unclear which information the active party provides to the passive party for it to carry out the IP leakage attack.

**Update:** The authors have clearly answered all my questions. However, I think the fact that the IP leakage issue is not actually related to the passive party dropping out is a major flaw in the presentation of the work and would require a major revision to fix. Therefore, I'm not upgrading my score.

**Questions:**

- Is the IP leakage issue related to passive drop-outs at all?
- Can you clarify if the "naive" alternative to robustness discussed in §5.1 is really impractical?
- How does the impact of drop-outs behave as a function on the number of passive parties? So is the presented accuracy drop also critical when one passive party in a, e.g., five-party setting drops out?
- Why is the evaluation in Table 3 only till p = 0.5; what happens in cases with high drop-out probability?

---

> ### Author Response · Authors · 2023-11-16
> **Authors' response**
>
> We thank the reviewer for the constructive reviews!
>
> We would like to thank the reviewer for the acknowledgement that the issues are clearly relevant in the deployment of VFL and that our method is effective. We thank the reviewer for the positive score of the paper.
>
> We would like to address the concerns raised by the reviewer here.
>
> **W1&Q1**. Our work is motivated by a realistic setting that the collaborating parties of a VFL system might quit the system due to the termination of the collaboration or network crash. Our paper is the first work to design a pluggable VFL framework that is robust to the unexpected quitting of passive parties in VFL. We establish that for existing VFL techniques, the quitting of passive parties causes two consequences: (1) a substantial performance drop; (2) intellectual property (IP) leakage through the passive party’s feature extractor. As the first paper to improve the robustness against unexpected quitting, we propose Party-wise Dropout and DIMIP to solve both two consequences, respectively. Even though they are different techniques, lacking either one of them cannot guarantee the robustness of VFL against the quitting of passive parties.
>
> **W2&Q2**. Thanks for the constructive comment. Notably, sometimes the passive party quits temporarily due to the network crash without any warning. The active party still needs to provide decent services after a party quits, but finetuning the head model requires time and causes communication overhead, and the low performance before the completion of the finetuning is unacceptable. We target that the active party can continue providing high-quality services after a party quits without any shut-down period. In addition, even if finetuning is acceptable, our framework can reduce the time and communication cost of finetuning with a higher start accuracy. As suggested by the reviewer, we conduct experiments on CIFAR100, and the results show that finetuning the head model of the vanilla VFL requires more than 15 epochs to achieve an accuracy higher than 34\%, while finetuning the head model of our method (p=0.1) only requires 3 epoch to achieve the same accuracy, which reduces the time cost significantly by 5 times.
>
> **W3.&Q3**. When there are more parties, our drop-out method will reduce the computational cost more compared with the multi-head training. To conduct multi-head training, the active party must compute all the terms in Equation 3, and the number of the terms in Equation 3 will increase rapidly when there are more parties. For the accuracy drop, it depends on the uniqueness and importance of the data provided by the passive party. It is reasonable that if a passive party’s data is more important, then the quitting of this party will cause worse performance degradation.
>
>
>
> We thank the reviewer for pointing it out and we will include the discussion in the final version.
>
>
>
> **W4**. We will refine Figure 1 in the final version. The active party will not “actively” provide any information to the passive party, and the feature extractor held by the passive party that was trained collaboratively with the active party’s labels is the only requirement for the passive party to conduct an IP leakage attack.
>
>
>
> **Q4**. It is unusual for the active party to rely more on the passive party than itself (I.e., by setting $p>0.5$). When $p$ is larger than 0.5, the accuracy before the passive party quits will converge to the accuracy of the Standalone accuracy of the passive party (i.e., p=1). However, we do not think it is a common case to set $p$ larger than 0.5.

---

> > ### Comment · Reviewer_ccWW · 2023-11-16
> > **Response to Rebuttal**
> >
> > Thanks for commenting on my concerns and questions, especially for providing the new experimental results.
> >
> > Re W1/Q1, my question remains if the IP leakage is strictly related to the passive party dropping out or if the same leakage could occur while that party is still online. This was not clearly answered.

---

> > > ### Author Response · Authors · 2023-11-16
> > > **Response to W1/Q1**
> > >
> > > Thanks for the reviewer's reply. Technically, the passive party does not have to conduct the attack after dropping out, and the same leakage could occur while the party is still online. The goal of our system is to make the passive party pluggable, which means that the active party does not need to put much additional effort into enhancing privacy and preserving the performance when the passive party quits. Thus, even though the active party's label IP leakage is not technically caused by the unexpected quitting, such leakage is the potential problem that the active party has to consider when the passive party quits the collaboration.
> > >
> > > We agree with the reviewer that such label leakage is not technically limited to quitting, and our defense can also protect the IP leakage when the passive party is still online. We will include this point in our discussion.

---

### Official Review · Reviewer_kTBy · 2023-10-27

**Soundness:** 3 good
**Presentation:** 3 good
**Contribution:** 3 good
**Rating:** 5
**Confidence:** 3

**Summary:**

Federated Learning exists some technical issues that cause people to decide the trade-off between model performance and privacy issues. The paper presents solutions to alleviate the privacy loss while retaining better performances. In vertical federated learning (VFL), there are two vulnerabilities caused by unexpected quitting of passive parties in the deployment phase – severe performance drop and active party’s label leakage.

On the one hand, the paper presents a VFL framework (PlugVFL) that could preserve the VFL model’s performance against unexpected exit of passive parties. By omitting the representations from the passive party with a certain probability p in each communication round (iteration of training in VFL), the framework combines a weighted sum of the vanilla model training and model training without passive parties. Could think of the weighted method as the dropout method in traditional neural networks.

Given that in the deployment phase passive parties could still access the active model’s data using the feature extractor, the chance of leaking labels in the active model increases. In a large dataset that has important or classified information, labels could be viewed as Intellectual properties that need to be protected. So PlugVFL, on the other hand, presents a method that could minimize the mutual information during the training phase. By calculating the variational upper bound, and minimizing the parameters that is less than the variational upper bound could achieve the result that passive parties hold as little as possible label information from the active party.

The paper also conducts different experiments to evaluate their framework’s effectiveness. PlugVFL can improve the accuracy after a passive party’s exit by more than 8% on CIFAR10, compared to passive party quitting in the normal situation. PlugVFL also prevents the passive party from fine-tuning a classifier that outperforms random guess levels even using the entire labeled dataset with only less than 2% drop in the VFL model accuracy, outperforming baselines significantly.

**Strengths:**

Given the workflow of how PlugVFL is designed, there are strong contributions that are achieved by it. In terms of defense, the model minimizes the mutual information between the representations of the passive party and the true labels, formulating the defense into an adversarial training algorithm that jointly minimizes the variational MI upper bound and production loss. It also improved the accuracy after the passive party's exit by more than 8% on CIFAR10. At the same time, all previous work provides IP protection in VFL training, and they are the first on IP protection on deployment phase.

**Weaknesses:**

There are several weaknesses given the fact and result this paper had presented. The experiment data is based on CIFAR dataset, and only split into two parties, one as active party and the other as passive party. Less than 1%(400) of the labeled data to fine-tune the model to perform model completion attack, and achieve comparable accuracy (~1% drop) compared to classifiers trained with all data just in part 2. Only two parties, and only one experiment, so that complexity is not guaranteed in larger and more complex federated tasks. Also the accuracy solely on party 2's data is not high in the first place. According to standard classifier performance on the CIFAR dataset, PlugVFL’s performance is not that ideal. In the expectation formula for robustness of retain accuracy when pass parties drop, the probabilities of dropping is not explained reasonably, because the the some popular classifier is around 70% standalone but PlugVFL has relative low accuracy (~44.95%) compared to standards, and only split two parties, so lack of reason and fact of the generality of the model. Lastly, the paper mentions naive solutions for mitigating the accuracy drop, such as fine-tuning the head model after a passive party quits. To actually apply this method and get the accuracy result is more convincing, because what if this method is better than the method proposed by the paper.

**Questions:**

There are a few questions that need to be addressed/clarified. Firstly, the label is counted as IP protection in federated learning, and so valid, because the label has useful representation that the passive model can extract. The whole logic is that after passive parties normally will only have access to the representation extractors, which allow passive parties to fine-tune classifier heads with very few labeled data. But how are you going to define the difference of label type? Because different label data types could have different presentations, and how to ensure that could be generalized not just text labels. If the only representation you could extractor is a numeric matching, then it is not qualified for IP protection. Secondly, reduce co-adaptation of the hidden neurons of the head predictor, similar to dropout, but why setting p = zero, will you still have accuracy after party 2 quit? Based on your definition, in each communication, p will be zero and therefore making the model equivalent to calculating prediction performance.

---

> ### Author Response · Authors · 2023-11-16
> **Authors' response**
>
> We thank the reviewer for the benevolent reviews!
>
> We appreciate the reviewer thinks PlugVFL has **strong contributions** with good performance and the acknowledgement that we are the first to study the problem.
>
> We will answer the reviewer’s questions and concerns here one by one.
>
> We summarize the reviewer’s concerns into three major points.
>
> 1. Experiments are limited to two-party experiments on CIFAR datasets.
>
> Notably, different from HFL, VFL does not have many parties. We follow previous works [1,2,3] to conduct experiments in the two-party scenario, which is widely used in the VFL community.
>
> We appreciate the reviewer’s suggestion about the datasets. We conduct experiments on the UCI_credit_card dataset [4] for the prediction of the default of credit card clients. We follow FATE [2] to assign ten attributes to one party and the rest to the other party. Due to the limited time, we currently can only report the results of the accuracy preservation against unexpected quitting, which are shown below:
>
> |  | Accuracy before party 2 quits (%)  | Accuracy after party 2 quits (%)  |
> |----------|----------|----------|
> | Vanilla VFL     | 77.25 |74.22    |
> | Standalone | N/A | 75.73    |
> | Multi-head training | 76.57 | 75.24    |
> |Ours (p=0.1) | 77.45 | 76.26 |
> |Ours (p=0.3) | 77.64 | 76.34 |
> |Ours (p=0.5) | 77.47 | 76.61 |
>
> It is shown that our framework is also effective in preserving the accuracy against the unexpected quitting on categorical datasets. We will include more results of this dataset in the revision.
>
> 2. The accuracy is low compared to the standard CIFAR classification.
>
> Since in the VFL setting, each party only holds a portion of the data. Consequently, the accuracy is the result with only half of the CIFAR image on each party. This also follows the convention of previous works[1,2,3].
>
> 3. Fine-tuning the head model as baselines.
>
> Notably, the passive party sometimes quits temporarily due to the network crash without warning. The active party still needs to provide decent services after a party quits, but finetuning the head model requires time and causes communication overhead, and the low performance before the completion of the finetuning is unacceptable. We target that the active party can continue providing high-quality services after a party quits without any shut-down period. In addition, even if finetuning is acceptable, our framework can reduce the time and communication cost of finetuning with a higher start accuracy. We conduct experiments on CIFAR100, and the results show that finetuning the head model of the vanilla VFL requires more than 15 epochs to achieve an accuracy higher than 34\% while finetuning the head model of our method (p=0.5) only requires 1 epoch to achieve the same accuracy, which reduces the cost significantly.

---

> ### Author Response · Authors · 2023-11-16
> **Authors' response (part 2)**
>
> We answer the reviewer’s questions here:
>
> **Q1**. Thanks for the constructive suggestion. To prove the generalization of our defense to different label types, we derive a theoretical robustness guarantee for our defense:
>
> **Theorem 1**. Let $g_{\psi}$ parameterize $q_{\psi}$ in Eq. 5. Suppose the passive party optimizes an auxiliary model $f^m(y|h^2)$ to estimate $p(y|h^2)$. For any $f^m$, we have:
>
> $\frac{1}{N}\sum\limits_{i=1}^N \log f^m(y_i|H^2_i) < \frac{1}{N}\sum\limits_{i=1}^N \log p(y_i) + \epsilon, $
>
> where
>
> $\epsilon = {I_{vCLUB} {g_\psi}}(h^2;y) + KL(p(y|h^2)||g_\psi(y|h^2)). $
>
>
>
>
> Notably, this theorem does not set any constraints or assumptions on the type of label. Specifically, if the task of collaborative inference is classification, we have the following results:
>
> $\frac{1}{N}\sum\limits_{i=1}^N\text{CE}\left[f^m(H^2_i), y_i\right] > CE_{random} - \epsilon, $
>
>
>
> where CE denotes the cross-entropy loss, $\text{CE}_{random}$ is the cross-entropy loss of random guessing.
>
> Such a theorem can prove the effectiveness of our defense under all types of labels.
>
>
>
> **Q2**. p=0 serves as a baseline reference, which is equivalent to vanilla VFL. In our final implementation and results shown in Figure 6, p is set as 0.5. As written in Eq. 3, when party 2 quits, we use zero vectors as the features from party 2.
>
> [1] Yang Liu, Yan Kang, Xinwei Zhang, Liping Li, Yong Cheng, Tianjian Chen, Mingyi Hong, and Qiang Yang. A communication efficient collaborative learning framework for distributed features. arXiv preprint arXiv:1912.11187, 2019.
>
> [2] Yang Liu, Tao Fan, Tianjian Chen, Qian Xu, and Qiang Yang. Fate: An industrial grade platform for collaborative learning with data protection. The Journal of Machine Learning Research, 22(1):10320–10325, 2021.
>
> [3] Zou, Tianyuan, Yang Liu, Yan Kang, Wenhan Liu, Yuanqin He, Zhihao Yi, Qiang Yang, and Ya-Qin Zhang. "Defending batch-level label inference and replacement attacks in vertical federated learning." IEEE Transactions on Big Data (2022).
>
> [4] I-Cheng Yeh and Che-hui Lien. The comparisons of data mining techniques for the predictive accuracy of probability of default of credit card clients. Expert systems with applications, 36(2):2473–2480, 2009. 7

---

> > ### Author Response · Authors · 2023-11-21
> > **Authors' response (part 3)**
> >
> > Dear reviewer,
> >
> > We would like to provide more experiment results with more than two parties we got. Due to the limited time, we can only have a part of the results of UCI_credit_card with three parties.
> >
> > |  | Accuracy before any party quits (%)  | Accuracy after party 3 quits (%)  | Accuracy after party 2&3 quit (%)
> > |----------|----------|----------|----------|
> > | Vanilla VFL     | 77.18| 74.32| 72.73   |
> > |Ours (p=0.1) | 77.36|76.36|75.68 |
> > |Ours (p=0.3) | 77.50|76.50|76.02 |
> > |Ours (p=0.5) | 77.43|76.58|76.18 |
> >
> > It is shown that when there are more parties quitting, the accuracy drop will be more serious. Our method can still preserve the accuracy after quitting well. We will run the experiments with three parties on CIFAR datasets and include the results in the revised paper. We hope the results can solve your concern about the performance of our method in the scenarios with more parties.
> >
> > Best,
> > Authors

---

> ### Author Response · Authors · 2023-11-22
>
> Dear reviewer,
>
> The Rebuttal/Discussion phase will end today. Could you please read our rebuttal and provide any feedback? We have made a lot of effort to derive the theoretical and empirical results during the rebuttal. Thanks a lot!
>
> Regards,
>
> Authors

---

### Official Review · Reviewer_71Ut · 2023-10-30

**Soundness:** 3 good
**Presentation:** 3 good
**Contribution:** 3 good
**Rating:** 6
**Confidence:** 3

**Summary:**

This paper addresses two critical issues in the VFL scheme, namely (1) performance degradation after passive clients drop from the scheme, and (2) mitigating IP leakage. The former issue is a novel observation in this work, where the inference performance degrades when a passive client drops. To tackle these issues, the paper introduces the plugVFL scheme, which incorporates two regularizations during VFL training.

**Strengths:**

1. The paper is the first to identify a significant problem in VFL, showing that when a party drops, it negatively impacts the VFL scheme's performance.
2. It also highlights the risk of IP leakage from feature extractors, demonstrating how passive parties can use their feature extractors for model completion and gain advantages within the scheme.
3. The alternative training design is intuitive and the paper effectively addresses the overhead of calculating mutual information objectives.
4. The paper successfully demonstrates the effectiveness of both objectives in a two-party VFL scenario for CIFAR image classification tasks.
5. The paper is well-organized and easy to read.

**Weaknesses:**

1. The link between the IP leakage issue and the deployment-time passive party dropping is not clear. The paper seems to conflate the solution to two unrelated problems. The "Active Model Completion (AMC)" is based on training time, which contradicts the deployment-time scenario.
2. The discussion of party dropping only considers two parties with limited evidence (CIFAR-10). The impact on more parties should be explored.
3. The mitigation of party dropping seems to offer minimal advantages over multi-head training (as shown in Figure 3).
4. The paper states that IP leakage is due to label information, supporting the design of mutual information regularization. However, the connection between IP leakage and label information is not well-established. Section 3.3 suggests that IP leakage is primarily about successful model completion, not direct extraction of labels from the active party. A counter-example is that a feature extractor can be learned without label information (i.e., unsupervised learning).
5. The scalability of the alternative training for multi-party cases is questionable. For instance, in a 3-party scenario, the performance should remain consistent in multiple cases (both passive party drops or either one party drops). The equation in Section 3 could be split into four, increasing computation overhead significantly.
6. The hyperparameter p setting for alternative training is debatable. Estimating the probability of dropping during training is challenging. The paper mentions that setting a relatively small p-value can improve the robustness of VFL. However, does this imply that the scheme should always set a small p?
7. There are inaccuracies in some statements, such as the assumption that "the service provider cannot afford to shut down the service while fine-tuning." In reality, the server could have a backup model to replace the running one without service disruption. Additionally, the statement, "Without loss of generality, we formulate our PlugVFL framework in the two-party scenario," should ideally be demonstrated in more than two-party scenarios, enabling downgrading to two-party scenarios.

**Questions:**

Is there a naive way to mitigate performance degradation when a party drops? For example, could setting a mean vector value instead of zero help?

Image classifications like CIFAR-10/CIFAR-100 may not be ideal examples of VFL tasks.  Is this framework effective on categorical datasets?

---

> ### Author Response · Authors · 2023-11-16
> **Authors' response**
>
> We thank the reviewer for the accessible reviews!
>
> First, we would like to thank the reviewer for the acknowledgement on the significance of the problem we study in VFL and agrees that we successfully and efficiently tackles the problem. Besides, we are encouraged that the reviewer found our paper is well-organized and easy to read.
>
>
>
> We would like to address the concerns raised by the reviewer here. We summarize and group the reviewer’s concerns into the following three perspectives:
>
> **Scenarios the problem is defined in** Concerns W1, W3, W4, W7 can be broadly grouped in this perspective.
>
> Our work is motivated by a realistic setting that the collaborating parties of a VFL system might quit the system due to the termination of the collaboration or network crash. Our paper is the first work to design a pluggable VFL framework that is robust to the unexpected quitting of passive parties in VFL. We establish that for existing VFL techniques, the quitting of passive parties causes two consequences: (1) a substantial performance drop; (2) intellectual property (IP) leakage through the passive party’s feature extractor. As the first paper to improve the robustness against unexpected quitting, we propose Party-wise Dropout and DIMIP to solve both two consequences, respectively. Even though they are different techniques, lacking either one of them cannot guarantee the robustness of VFL against the quitting of passive parties.
>
> Specific response for **W1**. The AMC attack is employed by the passive party who quits from the collaboration at the inference stage to infer the label of the samples in the inference phase. AMC is very suitable for the passive party to acquire the label information with the feature extractor obtained from the collaboration in the deployment phase.
>
> Specific response for **W4**. In the VFL scenario, usually it is a collaboration between two big companies, where one of the companies (referred to as the active party) spends money, labor and their own techniques to annotate the data to acquire the label information. Hence, the label information represents the intellectual property of the active party. Unsupervised learning is applicable when the data have enough information infer the label. However, in VFL settings, the data of one party do not contain enough information to train a well performed feature extractor independently, which is also the motivation of VFL. Thus, the feature extractor trained in VFL will contain the label information, which is the important IP of the active party.
>
> Specific response for **W3** and **W7**. We are proposing methods with minimal costs. The multi-head training will introduce extra computational overhead, as discussed in section 5.1 second last paragraph. And training a backup model from scratch will introduce extra costs. For reference purposes, to train a GPT-3 requires $450,000 in 2022, not to mention in VFL scenarios the communication costs are also tremendous.
>
> We will rephrase our statements accordingly.
>
> **Multi-party scenarios** Concerns **W2**, **W5** can be broadly grouped in this perspective.
>
> Notably, different from HFL, VFL does not have many parties. We follow previous works [1,2,3] to conduct experiments in the two-party scenario, which is widely used in the VFL community.
>
> Specific response for W5. In the implementation, we adopt a mask that randomly mask the features from passive parties, and we do not need to compute all the terms in Equation 3 for each iteration. Hence the computational overhead remains the same for different number of passive parties. The loss function in Equation 3 represents the expectation of the loss function we actually computes.
>
> **W6**. A small p leads to good joint performance while a large p leads to good robustness against unexpected quit. The selection should depend on the trade-off between joint performance and the trustworthiness of the collaborator. Our focus is to provide the companies in the VFL system a feasible solution to improve the robustness against the unexpected quitting. Estimating the p value for each party will be very interesting in real life, and it is not always about the technics. Sometimes it might also be related to the risk and cost control. For example, if a passive party will pay higher Liquidated Damages for the quitting, we should set a higher $p$ for this passive party, since they can pay more compensation for the loss of service degradation. Even though this is out of the scope of this paper, we will include this discussion in our revised paper.

---

> ### Author Response · Authors · 2023-11-16
> **Authors' response (part 2)**
>
> **Q1**. We thank the reviewer for the constructive suggestions on further baselines, Notably, it is not always feasible for the active party to collect the passive parties’ features and calculate the mean vectors due to privacy concern. However, we still conduct experiments to illustrate the accuracy of using the mean vectors on CIFAR10 and CIFAR100. The results are shown below:
>
> Cifar10:
>
> Accuracy before quitting: 74.62\%   Accuracy after quitting using zero vectors: 53.04\%  Accuracy after quitting using mean vectors: 52.19\%
>
> Cifar100:
>
> Accuracy before quitting: 44.95\%   Accuracy after quitting using zero vectors: 26.65\%  Accuracy after quitting using mean vectors: 25.33\%
>
>
>
> It is shown that using the mean vectors does not improve the accuracy after quitting, and the accuracy is even lower. We analyze that the reason is the mean vectors contain information of the passive party, but such information does not correspond to the test samples and might confuse the classifier on the active party.
>
>
>
> **Q2**. We appreciate the reviewer’s suggestion about the datasets. We conduct experiments on the UCI_credit_card dataset [4] for the prediction of the default of credit card clients. We follow FATE [2] to assign ten attributes to one party and the rest to the other party. Due to the limited time, we currently can only report the results of the accuracy preservation against unexpected quitting, which are shown below:
>
> |  | Accuracy before party 2 quits (%)  | Accuracy after party 2 quits (%)  |
> |----------|----------|----------|
> | Vanilla VFL     | 77.25 |74.22    |
> | Standalone | N/A | 75.73    |
> | Multi-head training | 76.57 | 75.24    |
> |Ours (p=0.1) | 77.45 | 76.26 |
> |Ours (p=0.3) | 77.64 | 76.34 |
> |Ours (p=0.5) | 77.47 | 76.61 |
>
> It is shown that our framework is also effective in preserving the accuracy against the unexpected quitting on categorical datasets. We will include more results of this dataset in the revision.
>
> [1] Yang Liu, Yan Kang, Xinwei Zhang, Liping Li, Yong Cheng, Tianjian Chen, Mingyi Hong, and Qiang Yang. A communication efficient collaborative learning framework for distributed features. arXiv preprint arXiv:1912.11187, 2019.
>
> [2] Yang Liu, Tao Fan, Tianjian Chen, Qian Xu, and Qiang Yang. Fate: An industrial grade platform for collaborative learning with data protection. The Journal of Machine Learning Research, 22(1):10320–10325, 2021.
>
> [3] Zou, Tianyuan, Yang Liu, Yan Kang, Wenhan Liu, Yuanqin He, Zhihao Yi, Qiang Yang, and Ya-Qin Zhang. "Defending batch-level label inference and replacement attacks in vertical federated learning." IEEE Transactions on Big Data (2022).
>
> [4] I-Cheng Yeh and Che-hui Lien. The comparisons of data mining techniques for the predictive accuracy of probability of default of credit card clients. Expert systems with applications, 36(2):2473–2480, 2009. 7

---

> > ### Comment · Reviewer_71Ut · 2023-11-20
> > **Thanks for the response**
> >
> > Thanks to the authors for addressing my concern. I will increase my score.
> >
> > Note there are some limitations:
> > (1) As shown in `UCI_credit_card` dataset, the impact of user dropping is not that significant.
> > (2) Lack of evidence in multi-party cases (also raised by other reviewers).

---

> > > ### Author Response · Authors · 2023-11-21
> > > **Thanks for the reply**
> > >
> > > Dear reviewer,
> > >
> > > Thanks for your reply and the valuable feedback after reviewing our rebuttal. Your insightful comments and suggestions helped in enhancing the quality of the paper. We are glad to reply to the two limitations you raised.
> > >
> > > (1) Compared with the CIFAR image datasets, the UCI_credit_card dataset is more simple. An accuracy drop of 3% can be thought of as significant on such a simple dataset. The results prove that our method can mitigate such an accuracy drop caused by the unexpected quitting on the tabular dataset. We can involve the evaluations on more tabular datasets in future works.
> > >
> > > (2) For the defense, we derive a theoretical result without the constraint of the number of parties in the system.
> > >
> > > **Theorem 1**. Let $g_{\psi}$ parameterize $q_{\psi}$ in Eq. 5. Suppose the passive party $k$ optimizes an auxiliary model $f^m(y|h^k)$ to estimate $p(y|h^k)$. For any $f^m$, we have:
> > >
> > > $\frac{1}{N}\sum\limits_{i=1}^N \log f^m(y_i|H^k_i) < \frac{1}{N}\sum\limits_{i=1}^N \log p(y_i) + \epsilon, $
> > >
> > > where
> > >
> > > $\epsilon = {I_{vCLUB} {g_\psi}}(h^k;y) + KL(p(y|h^k)||g_\psi(y|h^k)). $
> > >
> > >
> > >
> > >
> > > Notably, this theorem does not set any constraints or assumptions on the type of label. Specifically, if the task of collaborative inference is classification, we have the following results:
> > >
> > > $\frac{1}{N}\sum\limits_{i=1}^N\text{CE}\left[f^m(H^k_i), y_i\right] > CE_{random} - \epsilon, $
> > >
> > >
> > >
> > > where CE denotes the cross-entropy loss, $\text{CE}_{random}$ is the cross-entropy loss of random guessing.
> > >
> > > For the performance preservation against unexpected quitting, we conduct experiments with more than two parties. Due to the limited time, we can only have a part of the results of UCI_credit_card with three parties.
> > >
> > > |  | Accuracy before any party quits (%)  | Accuracy after party 3 quits (%)  | Accuracy after party 2&3 quit (%)
> > > |----------|----------|----------|----------|
> > > | Vanilla VFL     | 77.18| 74.32| 72.73   |
> > > |Ours (p=0.1) | 77.36|76.36|75.68 |
> > > |Ours (p=0.3) | 77.50|76.50|76.02 |
> > > |Ours (p=0.5) | 77.43|76.58|76.18 |
> > >
> > > It is shown that when there are more parties quitting, the accuracy drop will be more serious. Our method can still preserve the accuracy after quitting well. We will run the experiments with three parties on CIFAR datasets and include the results in the revised paper.
> > >
> > > Thanks,
> > >
> > > Authors

---

### Official Review · Reviewer_T9tD · 2023-10-31

**Soundness:** 2 fair
**Presentation:** 1 poor
**Contribution:** 2 fair
**Rating:** 3
**Confidence:** 4

**Summary:**

This paper designs a PlugVFL framework to address two problems caused by the unexpected quit of passive parties, i.e. severe performance degradation and intellectual property leakage of the active party’s labels.

**Strengths:**

The concerned problem is meaningful.

**Weaknesses:**

The experiments are not well designed.

**Questions:**

1. Fig. 3 and 6 plot the relationships between the test acc before party 2 quit and after party 2 quit, while Fig. 4 and 5 plot the relationships between the test acc and attack acc. Do they share the same hyper-paramters? What do the lines (which connect the points) mean? I donot get the meanings and reasons of the figures.

2. From Table 3, PlugVFL shows worse results than simply training independently. So does the proposed method make sense?

---

> ### Author Response · Authors · 2023-11-16
> **Authors' response**
>
> We thank the reviewer for the time devoted to the review.
>
> Firstly, we would like to thank the reviewer for the acknowledgement of the meaningfulness of the problem we are studying. It is encouraging! However, we politely disagree with the reviewer’s comments that our experiments are not well designed, and we answer the reviewer's questions one by one.
>
> Q1. Figure 3 plots the relationship between test acc before party 2 quits and after party 2 quits. As stated in the title of section 5.1, these two figures are measuring the accuracy drops (also known as the performance preservation ability). Stated in the second sentence in section 5.1, lines demonstrate the trends of the trade-off between accuracy before and accuracy after the quit with different level of defense. The best trade-off is achieved at the top-right corner.
>
> Meanwhile, Figures 4 and 5 measure the defense ability against label leakage, as stated in the title of section 5.2. In the first sentence of section 5.2, we specify the changes in the hyper parameter p=0, while all other hyperparameter remains the same and detailed in the Hyperparameter configurations. paragraph in section 5. As stated in the second sentence in section 5.2, Figures 4 and 5 measure the trade-off between attack accuracy and test accuracy under different level defense. The Evaluation metrics. paragraph specifies our definition of the evaluation metrics.
>
> Figure 6 is results for the joint performance under the best trade-off we observed from previous sections, as stated in section 5.3, the third sentence.
>
>
>
> Q2. When training independently, the active party achieves slightly higher accuracy compared with the accuracy of VFL after quitting. However, independent training does not allow the active party to utilize the passive party’s data and suffers from low accuracy before the passive party quits, which degrades the accuracy from 5% to 10%. Notably, it is also usually impractical and impossible to train independently. The company that deploys the system cannot afford the cost to shut down the service and retrain the model. The above discussion can found in last two paragraphs in section 5.1.

---

> > ### Comment · Reviewer_T9tD · 2023-11-23
> > **Thanks for the reponse**
> >
> > Thanks for the response!
> > However, my concerns are not fully addressed:
> > 1. As for the meaning of the figures, I don't think it is proper to connect the points (obtained from the experiments) with straight lines. Taking Figure 3 as an example, it plots the relationship between test acc before party 2 quits and after party 2 quits, but it is not guaranteed or a consensus that the relationship varies continuously (different from the acc respect to the epoch number). Therefore, simply comparing the lines between methods is not reasonable, especially the points are test with arbitrary parameters (the presented improvements may be produced by good luck).
> > 2. As you saied: "When training independently, the active party achieves slightly higher accuracy compared with the accuracy of VFL after quitting. ". Why not use the independent model instead of the proposed model when some parties quit.

---

### Meta-Review · Area_Chair_tPbi · 2023-12-10

**Metareview:**

This paper studies the scenario that, in vFL, when some of the passive parties quit the collaboration in the deployment of vFL. The proposed method tries to address the potential performance degradation and intellectual property (IP) leakage from the active party's labels. Overall the sentiment of the reviewers is lukewarm.

Strengths:
1. The problem studied is interesting and important.
2. Experiments are comprehensive.

Weakness:
1. The proposed solution might not have too much value. As one of the reviewers out, the training independently would be better. In that case, if passive party quits, why not just using the backup model (trained independently) as the serving model? Passive party quitting should be considered a critical issue in service, and I see having a backup model would be a better choice, especially this backup can be better both in performance and privacy.
2. As also pointed out by two reviewers, this paper mixed two issues, IP leakage + passive party quitting, and these two issues are not necessarily highly related. Rewriting the paper would take some substantial effort.

**Justification For Why Not Higher Score:**

The problem studied is interesting but the methodology and results are not very convincing.

**Justification For Why Not Lower Score:**

N/A

---

### Decision · Program_Chairs · 2024-01-16

Reject